# Carbon Emissions of the Tourism Telecoupling System: Theoretical Framework, Model Specification and Synthesis Effects

**DOI:** 10.3390/ijerph19105984

**Published:** 2022-05-14

**Authors:** Xiaofang Duan, Jinhe Zhang, Ping Sun, Honglei Zhang, Chang Wang, Ya-Yen Sun, Manfred Lenzen, Arunima Malik, Shanshan Cao, Yue Kan

**Affiliations:** 1School of Geography and Ocean Science, Nanjing University, Nanjing 210023, China; xiaofangduan@163.com (X.D.); zhanghonglei@nju.edu.cn (H.Z.); wangchang@nju.edu.cn (C.W.); njcss163@163.com (S.C.); kanyue116@163.com (Y.K.); 2Huangshan Park Ecosystem Observation and Research Station, Ministry of Education, Huangshan 245000, China; 3School of Management, Shandong University, Jinan 250100, China; 4School of Business, The University of Queensland, Brisbane 4072, Australia; y.sun@business.uq.edu.au; 5School of Physics, The University of Sydney, Sydney 2006, Australia; manfred.lenzen@sydney.edu.au (M.L.); arunima.malik@sydney.edu.au (A.M.)

**Keywords:** telecoupling, tourism telecoupling system, carbon emission, sending system, indirect spillover system, implied carbon trading

## Abstract

The flows of people and material attributed to international tourism exert a major impact on the global environment. Tourism carbon emissions is the main indicator in this context. However, previous studies focused on estimating the emissions of destinations, ignoring the embodied emissions in tourists’ origins and other areas. This study provides a comprehensive framework of a tourism telecoupling system. Taking China’s international tourism as an example, we estimate the carbon emissions of its tourism telecoupling system based on the Tourism Satellite Account and input–output model. We find that (1) the proposal of a tourism telecoupling system provides a new perspective for analyzing the carbon emissions of a tourism system. The sending system (origins) and indirect spillover system (resource suppliers) have been ignored in previous studies. (2) In the telecoupling system of China’s international tourism, the emission reduction effect of the sending system is significant. (3) The direct spillover system (transit) and indirect spillover system’s spatial transfer effects of environment responsibility are remarkable. (4) There is a large carbon trade implied in international tourism. This study makes us pay attention to the carbon emissions of tourists’ origins and the implied carbon trading in tourism flows.

## 1. Introduction

Tourism development is accompanied by large-scale, fast-growing and multiple flows of people, material, capital and service, which exerts major and complex impacts on the social economy and ecological environment from the local to the global [1,2]. From 2000 to 2018, the annual growth rate of global international tourism revenues was approximately 12%, reaching US $1.5 trillion, and as many as 1.4 billion tourists were involved [3,4]. International tourism growth continues to outpace the global economy. Meanwhile, the significant impacts of tourism development on ecological environment have also received great attention. In the context of people’s pursuit of a comfortable, fast and long-distance tourism, tourism is no longer a smokeless industry in our imagination. In fact, tourism sectors consume a lot of energy and emit vast quantities of greenhouse gases [5]. As a key indicator of the environmental impacts of tourism, carbon emissions have been widely considered and examined in multiple levels, such as global, national and regional [6,7,8].

However, previous work at different levels focused on tourism carbon emissions within a particular area, with little attention paid to tourism-integrated systems, and it especially overlooked that of tourism origins [9,10]. The long-distance flows of people, material and energy caused by international tourism exert profound impacts on global sustainability, which are the typical issues of telecoupling [7,11]. Liu et al. constructed a theoretical framework of telecoupling, which is defined as an umbrella concept that refers to the various socioeconomic and environmental interactions over distances [7]. It extends the concept of coupled human and natural systems in time and space and can help to explore the hidden elements in it [7,12,13]. Hence, the theoretical framework has gradually been applied in some fields, such as land use change [14,15], water resource management [16], ecosystem services [17], international trade [18,19] and biological diversity [20]. Tourism is also a typical phenomenon of remote human–environment interactions. The perspective of telecoupling can help to explore the hidden elements (such as the diverse influences on tourism origins and related regions) in tourism systems [7]. However, insufficient attention is paid to telecoupling in tourism, not to mention that in tourism environment effects.

Given the above considerations, this paper intends to explore the issues of tourism telecoupling in the dimension of carbon emissions. The specific aims of this study are to (1) construct a theoretical framework of the tourism telecoupling system, (2) establish the models of tourism carbon emissions from the perspective of a telecoupling system, and (3) taking China as the example, explore the synthesis effects of international tourism carbon emissions under this framework. There are two main contributions of this study. (1) First, it builds the theoretical framework of tourism telecoupling system and constructs the calculation model of tourism telecoupling system emissions, which expand the perspective and content of tourism system research. (2) Two important variables in tourism carbon emission system are identified, which are the emission reduction in the origins and the implied carbon trading in tourism flows. These help to improve the understanding of tourism system carbon emissions.

## 2. Literature

### 2.1. Tourism Carbon Emissions at Different Scales

Previous studies on tourism carbon emissions mainly focused on global, national and regional levels [21,22]. On the global scale, Gössling and the WTO-UNEP-WMO estimated the emissions originating from main tourism sectors (transport, accommodation and activities) based on inventory analysis. The results showed that the tourism sector accounted for approximately 3–5% of global carbon emissions in 2003 and 2005 [21,22]. Then, input–output models have emerged as a new macro-level approach to estimate tourism carbon emissions. Based on this method, the unexpected result is that the global carbon footprint of tourism was 4.5 GtCO_2_e in 2013, which is four times greater than that previously reported [9]. The research of the global scale is important not only to improve the understanding of tourism carbon emissions but also to formulate the global emission reduction target because of the issues with long-distance international transport [23,24].

On the national level, top–down and bottom–up approaches were the most commonly used [25]. The research perspectives of this level are manifold. Becken and Patterson [25] calculated CO_2_e emissions by analyzing tourism energy consumption and tourist behavior in New Zealand. Cadarso et al. estimated the contributions of residents and visitors to the tourism’s carbon footprint in Spain, and Meng et al. quantified the direct and indirect CO_2_ emissions of tourism of China [26,27]. In general, targeting the measurement of carbon emissions of tourism at the national level, existing research is mostly based on two theories: life cycle assessment and input–output, which essentially aim to measure carbon emissions from the perspectives of “consumption” and “production”. National-scale studies are the key to manage the emissions of tourism industry due to the huge size of domestic travel [28]. Some research have been conducted in the last decades, but the cases are mainly in developed countries [29,30,31,32,33,34,35] and there are no details of China’s international tourism using input-out method.

Moreover, the research on the regional level tends to pay attention to environmental vulnerability. Sun found that the island destination leads to a larger share of carbon emissions outside their geographic territory, and Farreny et al. highlighted the significant effects of transport on tourism carbon emissions in Antarctica; approximately 70% are attributable to cruising and 30% to flying [5,36]. These studies indicated that tourism greatly affects the sustainability of ecologically vulnerable areas, such as nature reserves and islands. However, these studies of tourism carbon emissions at different scales focus on a particular area, with little attention paid to tourism-integrated systems [37,38]. For example, the departure of tourists would reduce the carbon emissions of the origins. Tourism, especially international tourism, involves diverse and complex human–environmental interactions. The research of tourism-integrated systems should be conducted in the view of remote coupling, and the perspective of telecoupling could be an extension of global, national and regional scales.

### 2.2. Telecoupling

In recent decades, increased human interactions over long distances have profoundly impacted global socioeconomic and environmental sustainability [11]. Traditional one-place, one-way, or one-scale studies cannot analyze systematically the various social and environmental issues caused by international trade [39]. To address this challenge, Liu et al. [7] proposed the framework of the telecoupling system to explore the coupling mechanism considering multiple places and directions across time and space, which reveals the socioeconomic and environmental interactions between coupled human and natural systems over distances. The research framework is composed of five major, interrelated and structured components, which are system, flow, cause, agent and effect [7]. At the telecoupling system level, the framework includes a set of interacting coupled human and natural systems (according to the moving direction of flow, they are divided into three sub-systems: the sending system, receiving system and spillover system) through flows. At the sub-system level, every system consists of three interrelated components: agents, causes, and effects. This work expands the research scope of the interaction between remote areas and supplements the research program of complex problems with multi-scale and multi-system approaches [7,13]. Therefore, the telecoupling framework provides a more comprehensive explanation of distant human–environment interactions and makes key implications for global sustainability [13].

As a universal paradigm, the telecoupling system has been implemented in diverse fields, such as international trade [40,41], resource flow [42], species migration [43,44,45], and land use change [46,47]. These studies applied the telecoupling framework to explore the hidden elements within a single system and between multiple systems, and they provided some new explanations and made contributions to local and global sustainability management. In regard to tourism, Liu et al. [48] examined the telecoupled relationships between the Wolong Nature Reserve and the rest of the world, and they treated tourism as an input flow, which has a significant effect on the natural environment. Chung et al. [49] explored the relationships between biodiversity and nature-based tourism, with the conclusion that each 1% increase in biodiversity makes a 0.87% increase in tourism in a global context. Furthermore, Hulina et al. [45] applied the telecoupling framework to Kirtland’s warblers (a conservation-reliant migratory songbird) and analyzed the role of tourism and other human factors in the conservation of migratory species. However, the studies of tourism telecoupling, which mainly make tourism as a factor or tourism destination as an object, remain at the exploratory stage. The theory and some other key contents of tourism telecoupling framework are still unexplored.

## 3. Research Design

This study intends to discuss the tourism telecoupling system from the perspective of carbon emissions. China’s international tourism is adopted as an example to estimate the carbon emissions of the tourism telecoupling systems, which include the inbound tourism, outbound tourism and the whole sectors of international tourism. There are two reasons for choosing the international tourism of China as the case. First, international tourism, as a global activity, is a typical telecoupled phenomenon. Second, the international tourism of China has a large scale, and its partners are distributed worldwide. In recent years, the scale of the inbound and outbound tourism sectors in China has been among the highest in the world, and there exists a stable two-way tourists flow between China and the 189 international tourism partners. Hence, its representativeness and typicality are both significant.

### 3.1. Theoretical Framework of the Tourism Telecoupling System

The telecoupling framework provides a new perspective for the discussion of tourism system carbon emission. With the deepening of globalization and regional integration, the tourism industry in a given country is increasingly dependent on regions over distances, involving merchandise trade and personnel mobility. Therefore, the telecoupling under the tourism scenario needs to consider the resource supplier system, especially the island tourism destination, which is dependent a lot on the import of various resources. Therefore, this paper improves the telecoupling framework proposed by Liu et al. [7] in a tourism context and constructs the tourism telecoupling system, in which a resource supplier system is called the indirect spillover system. So, the tourism telecoupling system in this study includes the sending system, receiving system, direct spillover system and indirect spillover system (Figure 1) [50]. The different subsystems interact and influence each other through tourist flows, material flows, information flows and capital flows. Specifically, taking tourism carbon emissions as an example, the sending system is the tourist origin, where the residents leave for tourism motives (causes) and then reduce its resources consumption and carbon emissions. The receiving system is a certain destination, where tourists cause additional carbon emissions. The direct spillover system is the transit system, which leads to the main environmental issues in tourism sectors [9]. The indirect spillover system is the resource suppliers of the receiving system, which are indirectly affected by the tourism activities at destination. This tourism telecoupling system framework can not only discuss the environmental impact of tourism but also works in social, cultural and economic issues. In the international two-way tourism flow, the sending system is also a receiving system. Hence, this framework is systematic and dynamic.

### 3.2. Model Specification of Tourism Telecoupling System Carbon Emissions

In the context of tourism carbon emission, there is actually implied carbon trading in international two-way tourism flow. In the view of tourism telecoupling system, the reduction in the sending system carbon emission makes these synthesis effects more complex and more interesting. Therefore, the calculation of tourism telecoupling system carbon emissions contains six parts: the sending system, the receiving system, the direct spillover system and the indirect spillover system, the total carbon emissions of the tourism telecoupling system and the implied carbon trading of tourism. In this study, the carbon emissions are the CO_2_ emissions stemming from energy consumption [9,23,51]. The model specifications of every single part are described as follows.

(1) The sending system carbon emissions (*SS*). This part includes the CO_2_ emissions reduction in the sending system, resulting from residents’ departing for travel. Based on the statistics of the International Energy Agency (IEA), the daily CO_2_ emissions of residents include those of electricity and heat production, manufacturing industries and construction, transport, residential, commercial, and public services. The model of SS is as follows:(1)$$SS=T_{ij}\cdot EP_{i}\cdot t_{ij}$$
where i and j are the origin and destination, respectively, $T_{ij}$ is the arrival of the destination or the departure of the origin, $EP_{i}$ is the per capita daily CO_2_ emissions of tourists’ residences, including emissions of electricity and heat production, manufacturing industries and construction, transport, residential, commercial, and public services at residences from fossil fuels, and $t_{ij}$ is the tourists’ average length away from their origin. However, people generally leave their refrigerators on and even some other facilities causing stray currents in their homes, and this would make the results a little overestimated. From another point of view, the results would be a little underestimated, because the major international tourists are those with higher incomes, and their emissions per capita are above average. Therefore, it is considered as a reasonable estimate on the whole.

(2) The receiving system carbon emissions (*RS*). This includes the CO_2_ emissions generated by tourists at destinations. Studies on the carbon emissions of destinations are relatively mature, and the input–output approach is often used. So, this study adopts an environmentally extended input–output approach used in an interesting research made by Sun et al. [21]. In this method, detailed Tourism Satellite Account (TSA) is combined with global MRIO and CO_2_ emissions databases, thereby tracing the CO_2_ emissions originating from the consumption of tourist products and services. The model of *RS* is as follows [9,23,51]:(2)$$RS=q{\left(I-T{\hat{x}}^{-1}\right)}^{-1}\tilde{y}$$
where q is a 1 × **N** matrix of the carbon emissions intensity, ${\left(I-T{\hat{x}}^{-1}\right)}^{-1}$ is the Leontief inverse matrix, I is an **N** × **N** identity matrix, T  is an **N** × **N** MRIO matrix listing the international trade transactions between countries, $x=T1^{T}+y1^{y}$ is the total economic output, and y is an **N** × **M** matrix of the final demand of **M** global agents (households, governments, capital sector, and stocks) of **N** products. $\tilde{y}$ is an **N** × 1 matrix acting as the final demand block of the MRIO system.

(3) The direct spillover system carbon emissions (*DSS*). The *DSS* contains the emissions of transportation from the origins to the destinations, which is the key part of tourism emissions [23]. Referring to the previous literature, this study assumes that all international tourists travel by air [23]. The *DSS* model is as follows:(3)$$DSS=T_{ij}\cdot d_{ij}\cdot k_{ij}$$
where $T_{ij}$ is arrival of tourists to a destination, $d_{ij}$ is the actual round flight distance from the origin to the destination, and $k_{ij}$ is the emission factor per unit distance [23].

(4) The indirect spillover system carbon emissions (*ISS*). According to the statistics of the International Energy Agency, the energy industry (energy exploration) is the biggest contributor to carbon emissions of global trade. Hence, the *ISS* in this study is the CO_2_ emissions originating from fossil fuels exploitation by the resource supplier.
(4)$$ISS=EI_{j}\cdot m_{j}\cdot w_{j}$$
where $\;EI_{j}$ denotes the energy supply quantity of the resource supplier, $m_{j}$ denotes the carbon emissions per unit of energy extraction by the resource supplier, and $w_{j}$ denotes the proportion of the energy consumption of the tourism industry in the total energy consumption of the destination [23,52,53].

(5) The carbon emissions of the tourism telecoupling system (*TSS*). The *TSS* is the actual emissions of a certain tourism system. Hence, the *TSS* would be the sum of *RS*, *DSS*, and *ISS* minus *SS*.
(5)TTS=RS+DSS+ISS−SS

(6) The implied carbon trading of tourism (*ICT*). Two-way tourism flows between countries/regions not only include economic transaction but also imply spatial transfer of the environmental responsibility. Suppose two countries *A* and *B*; the two-way tourism flows make *A* and *B* both the sending system and receiving system. The *ICT* is the difference value between the change in the tourism carbon emissions of country *A* (the *RS_A_* minus the *SS_A_*) and the change in the tourism carbon emissions of country *B* (the *RS_B_* minus the *SS_B_*).
(6)$$ICT=\left(RS_{A}-SS_{A}\right)-\left(RS_{B}-SS_{B}\right)$$
where *RS_A_* is that when *A* is the receiving system, *SS_A_* is that when *A* is the sending system, *RS_B_* is that when *B* is the receiving system, and *SS_B_* is that when *B* is the sending system. When *ICT* > 0, *A* has an implied carbon trading surplus with *B*, and when *ICT* < 0, *A* has an implied carbon trading deficit with *B*. When *ICT* = 0, *A* has an implied carbon trading balance with *B*.

### 3.3. Data Resources

The data of this study come from multiple sources. The first part is tourism-related data. (1) The data of inbound and outbound tourists of China are from the World Tourism Organization. A spot of missing data is supplemented by the China Tourism Statistical Yearbook. (2) The data of tourism satellite accounts are from the World Tourism Organization, Organization for Economic Co-operation and Development, European Union, governmental reports, and journal articles. (3) The length of stay of inbound tourists is obtained from the Tourism Sample Survey of China, while that of outbound tourists is from online travel notes. More than 3000 notes from two major tourism websites (http://www.mafengwo.cn, accessed on 12 January 2020; https://www.ctrip.com, accessed on 12 January 2020) in China were captured and analyzed to ensure information saturation. (4) The distance from the origins to the destinations considers the actual distance of round direct flight route between their capitals. The emission factor per unit distance is from Gössling and Peeters [23]. The second part is energy-related data. (1) The energy supply quantity of the resource supplier is from the International Energy Agency. The daily carbon emissions of residents and the carbon emissions of energy extraction by the resource supplier are from the IEA annual report (http://data.IEA.org/, accessed on 10 January 2020), which is consistent with the calculation method of IPCC. (2) The tourism energy consumption coefficient is from a journal article [52]. The third part is economy-related data. Global multiregional input–output tables are from the EORA database (https://worldmrio.com/eora/, accessed on 10 January 2020). It should be noted that since the maintenance and updating of tourism satellite accounts in various countries around the world are not synchronized, the carbon emissions of China’s international tourism are estimated only in 2015.

## 4. Results

### 4.1. Inbound Tourism Telecoupling System

#### 4.1.1. Sending System

The CO_2_ emissions of the sending system reached 5.31 Mt in 2015; namely, the inbound tourists in China reduced the emissions of their origins by 5.31 MtCO_2_, accounting for 0.28% of the global residential carbon emissions in the same year [47]. Considering that this is only one country/destination, the figure is considerable. Moreover, the total and per capita CO_2_ emissions of the sending system show significant spatial heterogeneity. In terms of the total emissions, the United States ranked first, reaching 1.04 MtCO_2_, accounting for approximately one-fifth of the sending system emissions, which was followed by Korea (0.87 MtCO_2_) and Japan (0.43 MtCO_2_) (Figure 2). On a per capita basis, the emissions of developed countries are much higher than those of other countries. The per capita emissions of the United States (0.45 tCO_2_/capita), Canada (0.41 tCO_2_/capita) and Australia (0.36 tCO_2_/capita) are more than 20 times higher than those of developing countries, such as the Philippines (0.02 tCO_2_/capita), Cambodia (0.02 tCO_2_/capita) and Myanmar (0.02 tCO_2_/capita).

#### 4.1.2. Receiving System

The total carbon emissions of the receiving system are 17.01 Mt, i.e., inbound tourists produce 17.01 MtCO_2_ in China. On one hand, sending system emissions account for about 20% of that of receiving system. This shows that the estimations of the tourism system emissions in the past literature are too high. On another hand, the carbon emission in destinations is more than five times that of in origins. In other words, people’s carbon emissions from tourism activities are more than five times those from daily activities. Therefore, it is of great importance to encourage low-carbon tourism behavior. Regarding the total CO_2_ emissions of different origins, the total emissions of Pacific Rim countries are generally higher (Figure 3), with Korea (2.65 MtCO_2_), Japan (1.92 MtCO_2_) and the Russian Federation (1.46 MtCO_2_) occupying the top three countries. In per capita terms, the emissions of developed countries are also significantly higher than the others. For example, French emissions (0.72 tCO_2_/capita) are approximately 77 times higher than those of Myanmar (0.01 tCO_2_/capita).

#### 4.1.3. Direct Spillover System and Indirect Spillover System

The total CO_2_ emissions of the spillover system (direct spillover system and indirect spillover system) are 63.08 Mt. This part is the largest of the inbound tourism telecoupling system of China, and it is more than triple those of the receiving system (17.01 Mt). The spatial transfer of the environmental responsibility caused by tourism development cannot be ignored. In detail, the carbon emissions of the direct spillover system account for 61.38% (38.72 Mt). Moreover, the CO_2_ emissions of the direct spillover system (international transportation) are significantly higher than those of destinations, which is consistent with the previous studies [3,6,24]. On the other hand, the indirect spillover system emissions reach as high as 24.36 Mt, which shows the proposal of an indirect spillover system is necessary and important to improve the research on tourism system emissions.

### 4.2. Outbound Tourism Telecoupling System

#### 4.2.1. Sending System

In terms of outbound tourism, the sending system emissions reached 14.10 Mt in 2015. This shows that the outbound tourists reduce Chinese emissions by 14.10 MtCO_2_, which are approximately three times higher than the sending system emissions of inbound tourists. This accounts for 3.93% of the Chinese total residential carbon emissions in 2015 [51]. Specifically, as shown in Figure 4, Thailand (2.10 Mt), Japan (0.80 Mt) and United States (0.66 MtCO_2_) are the three largest countries that contributed to the sending system emissions of Chinese outbound tourism. Thailand is the preferred destination for Chinese residents to travel abroad in these years. On a per capita basis, Canada (0.23 tCO_2_), Australia (0.22 tCO_2_), United States (0.21 tCO_2_), Italy (0.20 tCO_2_) and France (0.19 tCO_2_) rank in the first five countries because of their longer distance from China and people’s longer stay duration. Meanwhile, those of Singapore, Korea and Japan are below half of them.

#### 4.2.2. Receiving System

The receiving system emissions reach as high as 21.60 Mt, which means Chinese tourists emitted 21.60 Mt CO_2_ in their destinations. This number is 1.53 times those saved in the sending system. Meanwhile, this ratio is 3.20 in Chinese inbound tourism. This shows that under the existing conditions, the energy consumption of Chinese residents’ tourism activities and daily activities is closer. In terms of specific countries/regions, as shown in Figure 5, Thailand (4.57 tCO_2_), Korea (2.64 tCO_2_), Singapore (1.38 tCO_2_), Vietnam (1.16 tCO_2_) and the United States (1.14 tCO_2_) are the five largest in the 189 destinations. The receiving system emissions of Thailand stand head and shoulders above others. The per capita emissions of the receiving system show similar characteristics with those of the sending system.

#### 4.2.3. Direct Spillover System and Indirect Spillover System

The carbon emissions of the spillover system (direct spillover system and indirect spillover system) of China’s outbound tourism are 96.97 Mt, of which the direct spillover system emissions reach 70.68 Mt, accounting for 72.89% and the indirect spillover system emissions are 26.29 Mt, accounting for 27.11%. Similar with that of inbound tourism, the spillover system emissions (both direct spillover system and indirect spillover system) are much larger than those of the sending system and receiving system. Meanwhile, an interesting indicator also reveals the particularity of China as a destination. The ratio of spillover system emissions and the receiving system emissions in China’s outbound tourism is 4.49, while that in China’s inbound tourism is 3.71. This shows that the spatial transfer of the environmental responsibility caused by tourism development of China is lower than that of other destinations as a whole. This result is consistent with the fact that the self-sufficiency of China’s tourism-related industries is relatively high.

### 4.3. Synthesis Effects of Tourism Telecoupling System Carbon Emissions

#### 4.3.1. Total Carbon Emissions of the Tourism Telecoupling System

The tourism telecoupling system emissions in this study are the actual emissions of China’s inbound and outbound tourism, which are the sum of the receiving system, the direct spillover system and the indirect spillover system minus the sending system. In 2015, the tourism telecoupling system emissions of China’s international tourism reached 179.24 MtCO_2_ (with the sending system at 19.41 MtCO_2_, the receiving system at 38.61 MtCO_2_, the direct spillover system at 109.40 MtCO_2_ and the indirect spillover system at 50.65 MtCO_2_). This result is consistent with the study of Sun et al., in which the carbon footprint of global tourism was discussed [9]. Regardless of the sending system and indirect spillover system, the direct spillover system is 2.83 times that of the receiving system and about 73.91% of the sum of the two. This result is in line with the cognition of the academic community; that is, the carbon emission of international transportation accounts for about 70% in international tourism [4,6,21]. The sending system emissions accounts for 50.27% of the receiving system, showing the importance of emissions reduction in the origins in the tourism system. Furthermore, the spillover system emissions are as high as 4.15 times those of the receiving system. The spatial transfer of the environmental responsibility in tourism cannot be ignored.

On a specific country basis, the carbon emissions of the tourism telecoupling system between China and another 189 countries showed a spatial heterogeneity. Geographical distance is the key variable of the telecoupling relationship. According to the geographical distance (GD) between the couple countries and the amount of their tourism telecoupling system emissions (TTSE), the 189 countries are divided into four categories: HL (high TTSE, long GD), HS (high TTSE, short GD), LL (low TTSE, long GD), and LS (low TTSE, short GD) (Figure 6). High TTSE means above the average values of the TTSE, and the low TTSE is the inverse. Long GD and short GD are in the same way. The bubble size is the number of inbound plus outbound. Figure 6 shows that the distribution basically conforms to Tobler’s First Law of Geography, except for the United States.

#### 4.3.2. Implied Carbon Trading in Tourism Flow

The implied carbon trading (*ICT*) of China’s international tourism is the net value of the change in the tourism carbon emissions of China (the *RS*_China_ minus the *SS*_China_) and the change in the tourism carbon emissions of another 189 countries (the *RS*_relevant countries_ minus the *SS*_relevant countries_). On the whole, China has a large carbon trading surplus in international tourism, reaching 23.26 CO_2_ Mt in 2015. This indicates that China exported 23.26 MtCO_2_ by two-way international tourism, accounting for 0.26% of the total CO_2_ emissions of fuel combustion in China [51]. Specifically, according to the definition of Section 3.1 above, China enjoys carbon trading surplus with 124 countries in international tourism, mainly including Thailand, Italy, France, Singapore, Malaysia, Spain, Cambodia, Vietnam, Indonesia, Germany, etc. Thailand is the largest one, as Figure 7 shows. On the other hand, China has a carbon trading deficit with 65 countries, including Russian, Japan, the United States, Mongolia, Korea, Canada, India, Philippines United Kingdom, Australia, etc. These amounts are almost half those of the former.

## 5. Discussion and Conclusions

With the deepening of globalization and the increasing of remote interaction between people and the environment, academia is forced to rethink the research limitations of traditional tourism system carbon emissions. The telecoupling theory provides a feasible path for more systematic and comprehensive research on the topic. Therefore, this paper explores the issues of tourism telecoupling from the perspective of carbon emissions by taking China’s international tourism as the example. The main findings are as follows: (1) the proposal of a tourism telecoupling system provides a new perspective for analyzing the carbon emissions of a tourism system. The sending system (origins) and indirect spillover system (resource suppliers) have been ignored in previous studies. (2) In the telecoupling system of China’s international tourism, the emission reduction effect of the sending system is significant. The sending system emissions accounts for 50.27% of the receiving system(destinations), while this part has been ignored in previous research and practice. (3) The direct spillover system (transit) and indirect spillover system’s spatial transfer effects of environment responsibility are significant. These two systems account for 89.29% of the total carbon emissions of the tourism telecoupling system. (4) There is a large implied carbon trade in international tourism. On the whole, China has a carbon-trading deficit in international tourism in 2015, reaching 23.26 MtCO_2_. Although there is no study that only estimates the carbon emissions of China’s outbound and inbound tourism, some relevant studies can be compared. In our results, the tourism telecoupling system emissions of China’s international tourism reached 179.24 MtCO_2_ in 2015. According to Lenzen et al. [9] and Meng et al. [27], about 480.2 and 208.4 MtCO_2_ have been generated from the domestic tourism, inbound tourism, consumption incurring domestically in relation to outbound travel in 2010. Both are larger than our results, which is mainly because it includes domestic tourism. At the same time, we can find there are some differences among some studies because of the different approaches and tourism system boundaries. For example, the emissions of tourism are 61.5 Mt in Australia and 1.438 Mt in New Zealand [54]. Of course, there are great discrepancies in the industrial level, natural conditions, stage of development, technical level and industrial structure among different countries.

This study has several theoretical and practical implications. First, the telecoupling system perspective is of benefit to the exploration of tourism systems. The tourism telecoupling system expands the perspective and content of tourism research. It is conducive to the in-depth exploration of the hidden elements in tourism systems by taking it as a continuously developing, globally expanding and dynamic complex system [48]. Furthermore, this framework is applicable not only to the environmental issues but also to the economic, social and cultural interactions among destination, origin and other related regions. This is also a supplement to Mill and Morrison’s tourism system model, Leiper’s tourism geographic system model, and Mckercher’s tourism complex system model [55,56]. Second, this study allows us to pay attention to the emission reduction effect of origins, the spatial spillover effect of tourism environment responsibility, and the implied carbon trading in tourist flows. Although these parts are significant in a tourism system, they have not received enough attention in the previous literature. The further discussion of these is necessary to expand and improve the tourism research. Third, in terms of practical implications, at the macro level, this study enables decision makers and managers to update their understanding of tourism system carbon emissions. The carbon emissions of a tourism system includes not only those of the destinations and the transportations but also those of the origins and resource suppliers. The strategies of tourism emissions reduction should be made from the perspective of global environmental governance. Finally, at the micro level, this study makes tourists realize the importance of low-carbon behavior for tourism emissions reduction. This study improves the macroscale understanding of governments and tourists regarding sustainable tourism. The sending system emissions account for 50.27% of the receiving system in China’s international tourism, which means people’s carbon emissions from tourism activities are about two times those of daily activities. Carbon taxes, low-carbon tourism industry, and responsible tourism behavior are all feasible attempts.

However, there are also some limitations. Although this study hopes to calculate the carbon emissions of a tourism telecoupling system as accurately as possible, the overall estimate may be slightly larger due to the slight differences in the classification of tourism satellite accounts among countries. Meanwhile, the tourism telecoupling system emissions could be explored in a long-time series. Not only the environment interactions but also the economic, social and cultural issues should be discussed. Furthermore, the exploration of the subsystem level of the tourism telecoupling system, such as agents, causes, and effects, should be performed to improve the effectiveness of policies targeting socioeconomic and environmental sustainability from local to global levels.

## Figures and Tables

**Figure 1 ijerph-19-05984-f001:**
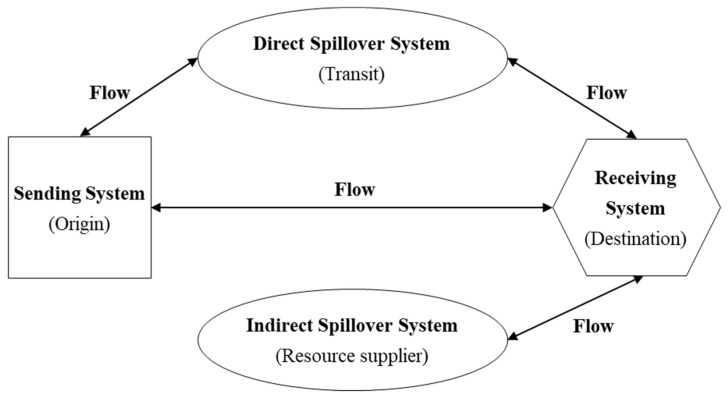
Framework of the tourism telecoupling system.

**Figure 2 ijerph-19-05984-f002:**
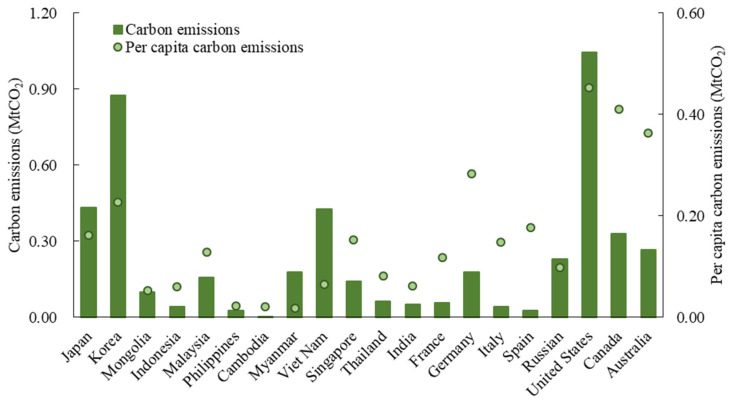
The sending system carbon emissions of China’s inbound tourism in 2015 (20 main countries are shown).

**Figure 3 ijerph-19-05984-f003:**
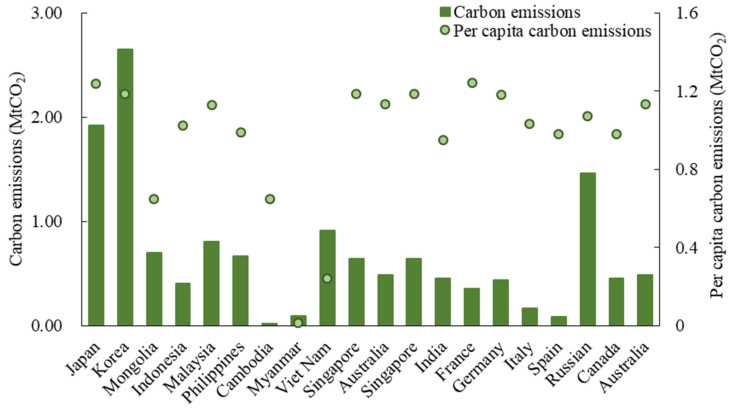
The receiving system carbon emissions of China’s inbound tourism in 2015 (20 main countries are shown).

**Figure 4 ijerph-19-05984-f004:**
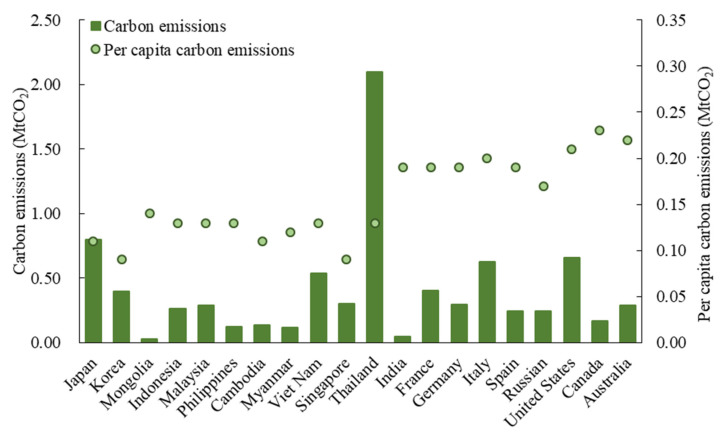
The sending system carbon emissions of China’s outbound tourism in 2015 (20 main countries are shown).

**Figure 5 ijerph-19-05984-f005:**
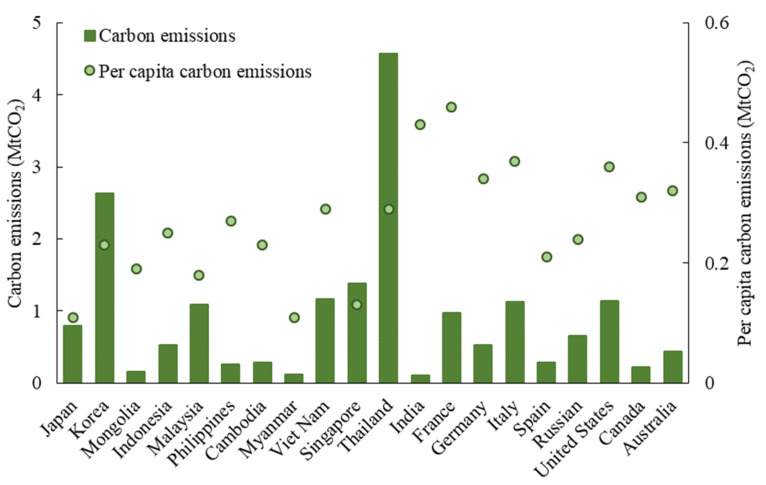
The receiving system carbon emissions of China’s outbound tourism in 2015 (20 main countries are shown).

**Figure 6 ijerph-19-05984-f006:**
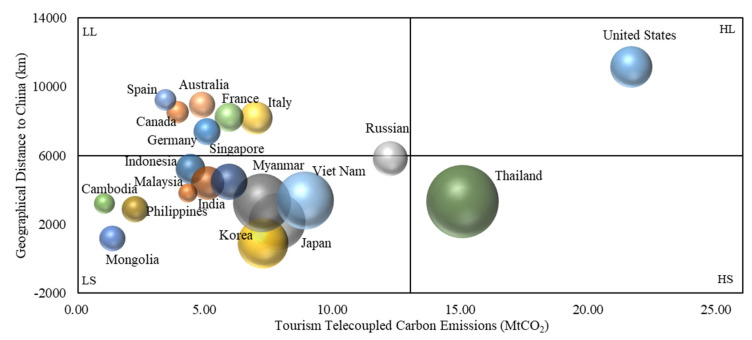
Geographical distance and carbon emissions of the tourism telecoupling system (20 main countries are shown).

**Figure 7 ijerph-19-05984-f007:**
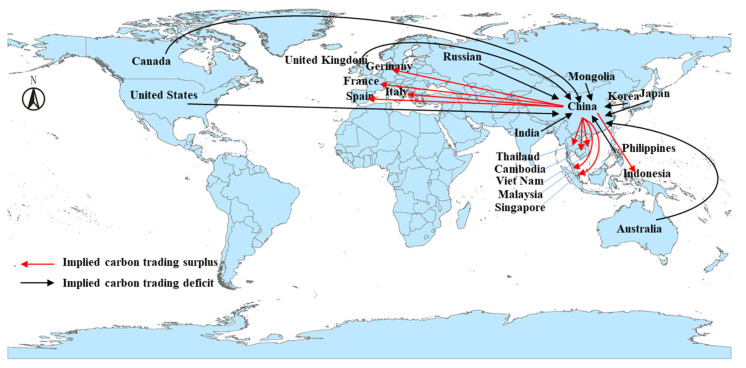
Implied carbon trading of international tourism in China (20 main countries are shown).

## Data Availability

The calculation data used in this paper come from the World Tourism Organization, Organization for Economic Co-operation and Development, and International Energy Agency, which have been explained in the main text.

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
