# Peer review of "Carbon Emissions of the Tourism Telecoupling System: Theoretical Framework, Model Specification and Synthesis Effects"

_ijerph, 2022, doi:10.3390/ijerph19105984_

Round 1

Reviewer 1 Report

The manuscript is clear, well structured and presented.
The description of the hypotheses and the design of the applied methodology are adequately explained.
The results are presented quite clearly.
A more detailed comparison with the literature results using different methodologies would be welcome.
The same goes for the convictions which could be more exhaustive through a more precise comparison with the literature data

Reviewer 2 Report

Dear authors,

I would like to congratulate you on your work. The topic is very relevant and contemporary, and is introduced in a simple and very readable way. However, some minor aspects still need to be improved, especially with regard to the model description and the graphical presentation of results. They are listed in the attached file.

Best regards,

Reviewer 3 Report

The paper is well prepared and I believe it will have contribution. However, the paper should be revised according to following suggestions before publication:

-The weakest part of the paper is about the Data. 

-Data used is 2015 which is old and not uptodate. There is no information about why 2015 data is used. If there is no new data, then it should be clarified and mentioned.

-Data sources and variables definitions are not well presented. In line 275: The tourism energy consumption coefficient is from journal articles 276 [24,48,50]. How it possible. 2 articles published on 2015 and other one published in 2002.

Literature review is weak and not uptodate. It should be enriched with few recent related papers

Results are well presented and discussed.

Round 2

Reviewer 3 Report

Paper can be accepted in present form.